# Equivalent Weight: Connecting Exoskeleton Effectiveness with Ergonomic Risk during Manual Material Handling

**DOI:** 10.3390/ijerph18052677

**Published:** 2021-03-07

**Authors:** Christian Di Natali, Giorgia Chini, Stefano Toxiri, Luigi Monica, Sara Anastasi, Francesco Draicchio, Darwin G. Caldwell, Jesús Ortiz

**Affiliations:** 1Advanced Robotics, Istituto Italiano di Tecnologia, Via Morego, 30, 16163 Genova, Italy; g.chini@inail.it (G.C.); Stefano.toxiri@iit.it (S.T.); darwin.caldwell@iit.it (D.G.C.); jesus.ortiz@iit.it (J.O.); 2Department of Technological Innovation and Safety Equipment, INAIL, 00169 Rome, Italy; l.monica@inail.it (L.M.); s.anastasi@inail.it (S.A.); 3 Department of Occupational and Environmental Medicine, Epidemiology and Hygiene, INAIL, Monte Porzio Catone, 00078 Rome, Italy; f.draicchio@inail.it

**Keywords:** biomechanical models—spine, job risk assessment, manual material handling, spine, low back, assistive technologies

## Abstract

Occupational exoskeletons are becoming a concrete solution to mitigate work-related musculoskeletal disorders associated with manual material handling activities. The rationale behind this study is to search for common ground for exoskeleton evaluators to engage in dialogue with corporate Health & Safety professionals while integrating exoskeletons with their workers. This study suggests an innovative interpretation of the effect of a lower-back assistive exoskeleton and related performances that are built on the benefit delivered through reduced activation of the erector spinae musculature. We introduce the concept of “equivalent weight” as the weight perceived by the wearer, and use this to explore the apparent reduced effort needed when assisted by the exoskeleton. Therefore, thanks to this assistance, the muscles experience a lower load. The results of the experimental testing on 12 subjects suggest a beneficial effect for the back that corresponds to an apparent reduction of the lifted weight by a factor of 
37.5%
 (the perceived weight of the handled objects is reduced by over a third). Finally, this analytical method introduces an innovative approach to quantify the ergonomic benefit introduced by the exoskeletons’ assistance. This aims to assess the ergonomic risk to support the adoption of exoskeletons in the workplace.

## 1. Introduction

There is evidence that work-related musculoskeletal disorders (WMSDs) are one of the major workplace health problems [1,2,3]. WMSDs include various pain and discomfort conditions in the muscles of the body; therefore, they are often grouped by the site of the pain and discomfort. For example, back pain, when the pain or discomfort is around the spine and supporting muscles and ligaments; upper limb disorders, comprising the shoulders, upper arms, forearms, hands, or wrists; and lower limb pain, which includes the hips, legs, knees, ankles, and feet [4]. Among these three main types of WMSDs, back pain is the main cause of activity limitation, work absenteeism, and lost productivity throughout much of the industrialized world [5,6,7,8]. More specifically, recent studies indicate that work-related low back disorders account for between 
26%
 and 
50%
 of the total number of reported cases of occupational WMSDs [9,10]. Moreover, a European report shows that based on the EU Labour Force Survey ad hoc modules, reported rates of the WMSDs across the EU generally increased from 
54.2%
 to 
60.1%
 between 2007 and 2013 [4]. This increase is not consistent across all EU countries: 14 countries (out of 28) have experienced an increase over this period. Nonetheless, the prevalence of WMSDs is still high in all the countries.

Prevalence of WMSDs is associated with manual material handling (MMH) [11,12]. Because of the high incidence of these disorders, their economic cost (e.g., days away from work and work compensation costs), and their impact on quality of life, many ergonomic interventions have been proposed, such as improving lifting technique, better foot positioning, and adjusting lifting height [13,14,15,16,17]. Nevertheless, these ergonomic solutions are not always applicable or sufficient. In this context, the use of exoskeletons in the workplace has attracted great interest.

Industrial (or occupational) exoskeletons are wearable devices that aim to reduce the exposure to risk factors for WMSDs. Among occupational exoskeletons, back support devices are specifically made for MMH. They aim to reduce the compressive load on the spinal discs and the associated risk of injury. They do this by producing/absorbing part of the torques normally generated by back muscles when a worker is lifting a load [18,19]. Recently, assessments of the effectiveness of such exoskeletons have been carried out worldwide by several groups [20,21,22]. Across these studies, different categories of metrics have been found useful to evaluate exoskeletons experimentally. Human biomechanics includes the analysis of movements and forces at play. For instance, estimated reduction in the compression forces and extension moments at the L5/S1 joint were found in [23]. Lifting behavior, such as trunk inclination and movement speed, is also of interest. In this respect, results focusing on different parameters have been described in the literature [23,24,25]. Electromyography is a technique very commonly used to assess exoskeletons, providing information on the activity of relevant muscles. Reductions of between 10% and 40% have been reported in the overall activation of lower back muscles (erector spinae), but muscles in different body areas may be of interest as well [26,27,28,29]. Physiological parameters (such as heart rate, metabolic cost, and breathing frequency) are indicators of global fatigue [30]. Mixed results in terms of metabolic cost can be found in the literature [31,32]. Exoskeleton-related parameters include data related to the assistance delivered by the exoskeleton, such as torque and power [33]. These can influence the effectiveness of a device as measured using the metrics mentioned above. Finally, questionnaires provide insights into the subjective perception of the exoskeleton, in terms of comfort, ease of use, and effectiveness [34,35]. As reported in [20,22], a consensus on the methods and metrics for the evaluation of occupational exoskeletons is still lacking, underlining that testing conditions and the performance metrics vary across the many available studies.

These evaluation methods and performance metrics are different from common practices used by companies to assess the ergonomic risk in repetitive movements and MMH activities within their workforce. Techniques that have been proposed for quantifying the amount of discomfort and postural stress caused by different body postures and job activities can be divided into observational and instrument-based techniques [36,37]. In the observational techniques, the angular deviation of a body segment from the neutral position is obtained using visual perception, while instrument-based techniques rely on devices attached to a person to record body posture. The observational techniques [38] include the Static Working Postures [39,40], the NIOSH Lifting Index [11], OUBPS [41], OCRA [42], HAL [43], and Strain Index [44]. These methods are used to investigate specific body areas and work activities. They are validated and included in internationally recognized standards (ISO 11228), but they also have some limitations such as evaluator-dependent results, poor correlation with the direct measurements of the physiological parameters, etc.

Commonly, the ergonomic risk of back disorders in MMH activities from the biomechanical point of view is quantified using the NIOSH method (e.g., NIOSH Lifting Index [11] and the revised NIOSH lifting equation [1,45,46]). This method takes into account different aspects of the task (e.g., posture, frequency, load, etc.) and it is widely accepted as the standard method to identify the risk associated with the vertebral column. Moreover, the popularity of using this method among field safety professionals is attributed to the simplicity and quantitative nature of the measurements: a weight scale, a tape measure, a goniometer, and a stopwatch can be used to determine the lifting task variables required to calculate the NIOSH Lifting Index. Among other methods can be also mentioned Continuous Assessment of Back Stress [47]. Note that there are some limitations with the NIOSH method as underlined in [48]. These are particularly associated with the complexity of occupational activities or the inapplicability of the NIOSH method due to particular postures, environmental conditions, sequential motions, or variable tasks [49,50].

A number of barriers are currently hindering the adoption of exoskeletons in the workplace. These include lack of consistency and comparability in reports describing available solutions and evidence of their effectiveness. This contributes to generally poor awareness and makes it difficult for interested adopters to find appropriate solutions for their needs. The growth in scientific literature in this area will certainly help all stakeholders and support informed adoption. Another important barrier to widespread adoption is the lack of viable methods to quantify the benefits of occupational exoskeletons in terms of the ergonomic risk. Commonly used risk indexes, such as the NIOSH Lifting Index, do not capture the effect of exoskeletons, making it difficult for Health and Safety professionals to rigorously assess existing solutions in terms that can be used to support adoption. This study addresses this barrier by investigating the effect and benefits of the use of an assistive back-support exoskeleton during MMH activities, by introducing an indirect measurement of its effectiveness in terms of apparent reduction of the lifted weight. The analytical method introduces an innovative quantification of the benefit of the exoskeleton usage by identifying a coefficient that can be taken into account when ergonomic risk evaluation is carried out. In particular, this study aims to lay the foundations of such an approach through quantitative measurements. The study analyzes the activity of trunk extensors and flexors, while holding four different loads and when subject to two different trunk forward static bending angles. This study is conducted, with and without, the exoskeleton to estimate the benefit in terms of muscle activity. The experimental testing aims to evaluate the variation of the muscle activities, and co-activation, for a population of six males and six females. This is done first without the exoskeleton (which will be considered the reference value) and then when using the exoskeleton to assist in supporting the external load.

## 2. Materials and Methods

### 2.1. Exoskeleton Prototype

The exoskeleton used in this study is named XoTrunk, shown in Figure 1. It is an active back-support exoskeleton designed to assist workers during manual material handling activities. XoTrunk is derived from the Robo-Mate active trunk prototype, presented in [51]. The device is worn similarly to a backpack, with the addition of articulated structures on the thighs. It provides physical assistance by contributing to the extension torque at the lower back and hips. The device consists of a main rigid frame at the back, which houses the on-board electronics. This is strapped onto the wearer at the shoulders and waist using custom-made soft braces. Two actuators are located on either side, approximately aligned with the hip joint. Each actuator consists of a brushless DC motor, a compact reduction gear, and a joint torque sensor. The actuators generate torques between the main back frame and the corresponding articulated leg link. Details on the actuators and their low-level control can be found in [52]. Each leg link is composed of articulated structures connected to the thigh strap at the distal end. Details on the kinematics and physical attachments on the XoTrunk are reported in [53]. The waist belt is designed to load most of the weight of the exoskeleton (7.5 kg) on the hips over the iliac crest [54].

The reference values for the actuator torques are generated so as to provide appropriate patterns of assistance when required, while keeping hindrance at a minimum. Ultimately, this maximizes the biomechanical effectiveness of the device and its acceptance by users. This approach to the control runs in real-time, using the on-board computer and joint level sensory readings, consists of (a) a high-level layer that detects the activity, e.g., walking, bending, lifting, etc. (further details are provided in [55]), and (b) a mid-level layer that adjusts the torque patterns as appropriate during each activity (see in [56] for more information). In the present study, the control algorithm was simplified to meet the requirements of the experimental protocol and the static nature of the physical tasks performed (more details in Section 3.1). Specifically, the exoskeleton was programmed to generate constant torques (24 Nm for women and 30 Nm for men) during the experimental task. For male participants, we selected 30 Nm torques, using the authors’ previous experience [56] to the static nature of the assessed task. For female participants, we used the same proportion found in the reference masses suggested by the ISO TR 12295:2014 (25 kg for men and 20 kg for women), and therefore selected 24 Nm constant torque for women. An upper limit constrained, constant torque value was used during the experimental section for safety reasons, as it prevents dangerous oscillations of the torque reference which could arise due to possible system noise.

### 2.2. Biomechanics of Load Handling

It is a well-established fact that the stresses induced at the low back during MMH are due to a combination of the weight lifted and the person’s technique for handling the load [11]. Specifically, the load held in the hands, as well as the person’s body masses, create moments at the various joints. The skeletal muscles exert forces on these joints to counteract the moments created by the load and body weight. From the mechanical stress standpoint, the muscles are unfavorably positioned, being very close to the vertebral column, as they act through relatively small moment arms. This means that they can produce large motions with small degrees of shortening, but for any external load operating on the body, high muscle and joint forces are produced.

The lumbar spine can be thought of as a set of small links separated by flexible articulations (discs). Complex models and specific physiologic data can be used to predict the whole kinematic analysis in each disc during specific lifting activities [11]. Because the clinical and biomechanical data indicate that the greatest problem is in the lower spine, the lumbosacral joint (the L5/S1 disc) has been used to represent the spinal stresses during lifting [57,58,59]. These models have clearly shown that during weight lifting, the bending moment at the lumbosacral joint can become quite large. To balance this moment, the muscles of the low back region (particularly the erector spinae group) must exert high forces due to their small moment arms (ranging from 38 to 50 mm). The high forces generated by the low back muscles are the primary source of compression forces on the lumbosacral disc. This is illustrated by the biomechanical model in Figure 2A for a person lifting a load. The biomechanical model considers the forces and moments operating at the L5/S1 disc while lifting a load. Considering the subject bending forward by an angle 
θ
, measured from the normal of the vertebral/discs interface (L5/S1) with respect to the vertical axis. The zero position corresponds to vertically upright and positive values indicate forward bending from this datum. 
$F_{B}$
 is the force due to the upper body weight (
UBW
) applied at the center of gravity of the upper body portion above the lumbosacral joint. 
$F_{W}$
 is the force generated by the load held by the person, which is the weight multiplied by the gravitational vector. Both 
$F_{B}$
 and 
$F_{W}$
 are located at 
$r_{B}=\left[r_{Bx},r_{Bz}\right]$
 and 
$r_{W}=\left[r_{Wx},r_{Wz}\right]$
 away from the application point of the normal compressing force (
$F_{C}$
) on the vertebral/discs interface (L5/S1). The L5/S1 vertebral/discs interface is considered as the origin of the Cartesian coordinate system. The line-of-action of the muscles of the lower lumbar back are assumed to generate the overall force 
$F_{M}$
 acting parallel to the normal compressing force on the vertebral/discs interface. It has a moment-arm (
$r_{M}$
) of 50 mm. For the completeness of the biomechanical model, the contribution of intra-abdominal pressure in relieving compression on the lumbar spine was experimentally estimated [57]. This procedure required correlating the subject-specific pressure data with the torque (
$T_{H}$
) and angle (
$A_{H}$
) estimated at the hip of the subjects involved in the experiments. Based on the results achieved in [57], the abdominal pressure is 
$P_{A}=10^{-4}\left(0.6516-0.005447\left(A_{H}\right)\right)T_{H}^{1.8}$
. The abdominal force function of the diaphragm area (
$D_{A}$
) is 
$F_{A}=P_{A}D_{A}$
. For the lifting position depicted in Figure 2A, it is estimated that the abdominal force (
$F_{A}$
) line of action is about 90mm anterior to the center of the lumbosacral disc. The force generated by the spinal muscles and the lumbar compression can be calculated by writing the free body diagram equilibrium for the biomechanical model displayed in Figure 2A:
(1)
$$T_{L5/S1}=F_{M}r_{M}+F_{A}r_{A}-\left(F_{B}r_{Bx}+F_{W}r_{Wx}\right)cos\theta -\left(F_{B}r_{Bz}+F_{W}r_{Wz}\right)sin\theta$$


(2)
$$F_{C}=F_{M}-F_{A}+\left(F_{B}+F_{W}\right)cos\theta$$

where Equation (Equation 1) represents the equilibrium of the moment about the L5/S1 joint, while Equation (Equation 2) is the force equilibrium of the upper-body acting on the upper spine segment above the L5/S1. Figure 2B shows the biomechanical model for a person lifting a generic load while assisted by a specific trunk-support exoskeleton that generates assistive forces perpendicular to the spine [60]. The exoskeleton generates a torque, 
$T_{X}$
, at the actuated joint, which consequently generates forces at the exoskeleton–body interfaces (i.e., waist belt, shoulder straps, and thigh braces). These forces—
$F_{X}$
, 
$F_{X1}$
, and 
$F_{X2}$
—displayed in Figure 2B are located at the shoulder straps (
$r_{X}$
 along the z axis of the spine), waist belt (
$r_{X1}$
 along the z axis of the spine), and at the thigh braces, respectively. Equations (Equation 3) and (Equation 4) show the moment and forces equilibrium at the exoskeleton joint and upper link, respectively. Equation (Equation 5) shows the equivalence of the assistive force acting at the shoulder strap as a function of the application radius and the given torque generated by the exoskeleton during the assistance.

(3)
$$T_{X}=F_{X1}r_{X1}-F_{X}r_{X}$$


(4)
$$T_{X}/r_{X2}=F_{X}-F_{X1}$$


(5)
$$F_{X}=\frac{T_{X}\left(1+r_{X1}/r_{X2}\right)}{r_{X1}-r_{X}}$$


As the physical weight of the exoskeleton is offloaded to the user’s waist through the belt, the exoskeleton weight contribution on the equivalences is not relevant and can be omitted from the calculation. The force generated by the spinal muscles and the lumbar compression while assisted by a back-support exoskeleton can be calculated by writing the free body diagram equilibrium for the biomechanical model displayed in Figure 2B: 
(6)
$$T_{L5/S1}=F_{M}^{*}r_{M}+F_{A}r_{A}+F_{X}r_{X}-\left(F_{B}r_{Bx}+F_{W}r_{Wx}-F_{wX}r_{wX}\right)cos\theta -\left(F_{B}r_{Bz}+F_{W}r_{Wz}\right)sin\theta$$


(7)
$$F_{C}^{*}=F_{M}^{*}-F_{A}+\left(F_{B}+F_{W}+F_{wX}\right)cos\theta$$

where 
$F_{M}^{*}$
 and 
$F_{C}^{*}$
 are the muscular and compression forces, respectively, modified with respect to Equations (Equation 1) and (Equation 2) by the use of an assistive exoskeleton. 
$F_{wX}$
 and 
$r_{wX}$
 account for the exoskeleton weight and its distance from the spine (about 10 cm). Note that the exoskeleton under analysis offers an assistive force (
$F_{X}$
) directed perpendicularly to the torso, and thus it does not contribute to compressing the lumbar spine. Therefore, the exoskeleton’s assistive contribution (
$F_{X}$
) does not appear in Equation (Equation 7). Recent studies [61,62] recorded the effort and the EMG activity of trunk muscles. The experimental tests allow the mapping of the EMG–force relationship. An ideal linear EMG–force relationship is observed for back muscles. Thus, this relationship can be simplified with a rate coefficient 
α
 that relates the envelope of the EMG of the erector spinae muscle with the generated low back force (
$F_{M}$
):
(8)
$$F_{M}=\alpha EMG_{ES}$$

where the 
$EMG_{ES}\left[\%\right]$
 is the envelope of the EMG of the erectus spinae normalized with respect to the maximum voluntary contraction (MVC). The relationship and specifically 
α
 is unknown and subject to variability due to the physical characteristics of the test subjects. Therefore, the coefficient of Equation (Equation 8) should be calibrated against individual maximum voluntary activation (MVA) [63]. Although there is a nonlinear relationship between the EMG measurements and muscle forces due to cross-talking effects [64,65], in [63], by applying regression analysis, a relationship was defined between the average peak reaction moment about the L5/S1 joint and the MVA of the erectus spinae. In this study, the subject performed the MVA while lying prone on a bench and positioned so that the antero-superior iliac spines were aligned with the edge of the bench. The upper body was unsupported off the end of the bench in a horizontal trunk position. For the MVA in extension, a dynamometer was used to measure the isometric back extension force of the subject. This research found a simple relationship between the MVA and the test subject’s body mass (
BM
), acting at the joint L5/S1. This is expressed in [Nm] as 
$MVA\left[Nm\right]=T_{L5/S1}^{MAX}=-119+4.78*BM\left[kg\right]$
 [63]. The latter equation verifies the following equilibrium: 
$T_{L5/S1}^{MAX}=F_{M}^{MAX}r_{M}-F_{B}r_{Bz}$
, where 
$T_{L5/S1}^{MAX}$
 is the maximum torque generated at the spine due to the specific task. The back muscles generated their maximum value of force as expressed in 
$F_{M}^{MAX}$
, and as the subject performed the MVA while lying prone on a bench, the load applied for the equilibrium is the only upper body weight expressed with 
$F_{B}r_{Bz}$
. Therefore, Equation (Equation 8) can be manipulated to give

(9)
$$F_{M}=\frac{T_{L5/S1}^{MAX}+F_{B}r_{Bz}}{r_{M}}EMG_{ES}\left[\%\right]=\frac{\left(4.78BM-119\right)+F_{B}r_{Bz}}{r_{M}}EMG_{ES}\left[\%\right]$$


Therefore, it is possible to quantify the effect of the exoskeleton on the compressing force by subtracting Equation (Equation 7) from Equation (Equation 2):
(10)
$$F_{C}-F_{C}^{*}=F_{M}-F_{M}^{*}-F_{wX}cos\theta$$


As expected, the exoskeleton generates a benefit in terms of muscular effort reduction, which becomes

(11)
$$F_{M}^{*}=\beta F_{M}$$

where (
1−β
) is the muscle force reduction. Thanks to the linear relationship between the EMG and relative generated low back forces, the reduction rate (
1−β
) could be directly evaluated by direct measurement and comparison of the 
$EMG_{ES}\left[\%\right]$
 with and without the use of the exoskeleton. Consequently, by applying the same hypothesis, the compressive force is also reduced by (
1−γ
), as follows:
(12)
$$F_{C}^{*}=\gamma F_{C}$$


Thanks to the use of the exoskeleton, the rate of compressing force reduction, varies in proportion to the constant coefficient 
γ
 as a function of both low back muscle activation and compressing force:
(13)
$$\gamma =1-\frac{\left(1-\beta \right)F_{M}-F_{wX}cos\theta }{F_{C}}$$


### 2.3. Equivalent Weight

As demonstrated previously when considering the biomechanical model, the use of an exoskeleton helps to reduce muscle activity of the erector spinae (ES) during the target activity of manual material handling. However, as both the compressive and muscular forces are not easily measured in a working environment, where such assistive device would be adopted and used, we aim to identify the benefits of exoskeleton use through a more concrete and direct approach. Moreover, to make the exoskeleton performance more understandable for a wide audience, this approach considers the effect and benefits that an exoskeleton can bring through the following example. The user, thanks to the assistance provided by the exoskeleton, often reports (anecdotally) a feeling of reduced stress at his/her back, and therefore a sensation of bearing a load lower than the real one. This apparent sensation provides the motivating hypothesis to quantify this impression. The study’s aim, thus, is to map the actual supported weight with the “equivalent weight” (EqW). The term EqW (
$F_{W}^{*}$
) is introduced and defined as follows: EqW is the perceived weight of the load held by the subject, which is apparently reduced by the assistance provided by the exoskeleton. In other words, the spine experiences loads lower than the actual 
$F_{W}$
, in terms of 
$F_{M}$
 and 
$F_{C}$
 in Equations (Equation 11) and (Equation 12).

We propose the formulation of a coefficient to be employed to complement occupational ergonomic risk assessment methods, taking into consideration the effect of exoskeletons. In other words, the EqW is an attempt to associate with the exoskeleton a numerical coefficient that quantifies its benefit for future ergonomic assessment methods. We derive the EqW in two different ways: (1) the first one is based on the biomechanical model analysis, while (2) the second method is based on interpolation and regression analysis of the muscular activity. Both methods are functions of the trunk extensor muscles activities and the weight lifted. The first method, starting from Equations (Equation 1) and (Equation 2), derives a set of equivalences that verify the reduction in the lifted weight and thereby determine a coefficient 
ψ
. The second method provides a coefficient 
σ
 (EqW reduction rate) by measuring the muscular activity, with and without the exoskeleton.

#### 2.3.1. Method 1: Biomechanical Model

This approach uses the hypothesis that 
$T_{L5/S1}$
 (of Equation (Equation 6)) is unchanged by the use of the exoskeleton: this hypothesis is valid if the exoskeleton does not alter the user’s movements and dynamics [23]. Consequently, Equations (Equation 6) and (Equation 7) can be modified by substituting the lifted weight (
$F_{W}$
) with the EqW (
$F_{W}^{*}$
). The low back muscular forces change accordingly (
$F_{M}^{*}$
), but the exoskeleton does not explicitly appear in the equivalence. The effect of the exoskeleton is taken into account through the reduction of the lifted weight (the EqW, 
$F_{W}^{*}$
). The equivalences are

(14)
$$T_{L5/S1}=F_{M}^{*}r_{M}+F_{A}r_{A}-\left(F_{B}r_{Bx}+F_{W}^{*}r_{Wx}-F_{wX}r_{wX}\right)cos\theta -\left(F_{B}r_{Bz}+F_{W}^{*}r_{Wz}\right)sin\theta$$


(15)
$$F_{C}^{*}=F_{M}^{*}-F_{A}+\left(F_{B}+F_{W}+F_{wX}\right)cos\theta$$


To quantify the effect of the exoskeleton on the EqW, Equations (Equation 1) and (Equation 14) are combined as follows: 
(16)
$$F_{M}r_{M}-F_{W}r_{Wx}cos\theta -F_{W}r_{Wz}sin\theta =F_{M}^{*}r_{M}-\left(F_{W}^{*}r_{Wx}-F_{wX}r_{wX}\right)cos\theta -F_{W}^{*}r_{Wz}sin\theta$$


The exoskeleton generates an apparent reduction of the lifted weight, which can be quantified as

(17)
$$F_{W}^{*}=\psi F_{W}$$


(
1−ψ
) is the lifted weight reduction. By substituting Equations (Equation 11) and (Equation 17) into Equation (Equation 16), the relationship between 
ψ
, as function of the lifted weight, and the muscular force is derived and given by

(18)
$$\psi =1+\frac{\left(\beta -1\right)F_{M}r_{M}+F_{wX}r_{wX}cos\theta }{F_{W}\left(r_{Wx}cos\theta +r_{Wz}sin\theta \right)}$$


By substituting Equation (Equation 9) into Equation (Equation 18), the following relationship for the EqW reduction rate as a function of the EMG of the ES muscular group (measured without the exoskeleton) and the actual lifted weight is obtained:
(19)
$$\psi =1+\frac{\left(F_{M}^{*}-F_{M}\right)r_{M}+F_{wX}r_{wX}cos\theta }{F_{W}\left(r_{Wx}cos\theta +r_{Wz}sin\theta \right)}=1+\frac{\left(\beta -1\right)\alpha EMG_{ES}r_{M}+F_{wX}r_{wX}cos\theta }{F_{W}\left(r_{Wx}cos\theta +r_{Wz}sin\theta \right)}$$


#### 2.3.2. Method 2: Data Interpolation

In the data interpolation method, the first step is to record the overall trunk extensor muscles activities, with and without the exoskeleton, and thus compute the selected activity index. Two plots (one at 
$0^{\circ }$
 inclination and the second at 
$30^{\circ }$
 inclination) can be drawn. These plots have as their y-axis the normalized value of EMG, with respect to the MVC. The x-axis shows the weight lifted. This method is validated on both populations of the study, showing separated trends for female and for male. The discrete values can be interpolated with linear functions, thus obtaining four functions as

(20)
$$g\left(x_{L}\right)=EMG_{\%}^{nX0}=a_{nX0}x_{L}+b_{nX0}$$


(21)
$$f\left(x_{L}\right)=EMG_{\%}^{X0}=a_{X0}x_{L}+b_{X0}$$


(22)
$$EMG_{\%}^{nX30}=a_{nX30}x_{L}+b_{nX30}$$


(23)
$$EMG_{\%}^{X30}=a_{X30}x_{L}+b_{X30}$$

where 
$a_{X30}$
 is the first-order coefficient of the condition with exoskeleton at a 
$30^{\circ }$
 inclination (the index “nX” is used for the modality without the exoskeleton, whereas the “0” at the index stands for the test recorded at 
$0^{\circ }$
 inclination), 
$b_{X30}$
 is the intercept of the same test condition (with exoskeleton at 
$30^{\circ }$
 inclination). 
$x_{L}$
 is the independent variable of the function, which corresponds to the lifted weight. To derive the EqW as a function of both trends, i.e., the muscular activation in the normal condition and the muscular activation while using the exoskeleton, we apply the inverse of a composite function. The composite function is 
$g^{-1}$
 (
$f(x_{L})$
). The inverse of the function 
$g(x_{L})$
 returns the lifted load, taking as input function the normalized muscular activity:
(24)
$$g^{-1}\left(EMG_{\%}^{nX0}\right)=\frac{EMG_{\%}^{nX0}-b_{nX0}}{a_{nX0}}=x_{L}$$


If considering inputting the latter function (Equation (Equation 24)) with the obtained muscular activity using the exoskeleton (
$EMG_{\%}^{X0}$
), the function returns the corresponding rated value of the lifted weight commensurate to the specific muscular activity. Thus, the EqW can be calculated by feeding the equation 
$g^{-1}$
 (Equation (Equation 24)) with the rated value of muscular activity with the exoskeleton:
(25)
$$g^{-1}\left(EMG_{\%}^{X0}\right)=\frac{EMG_{\%}^{X0}-b_{nX0}}{a_{nX0}}=\frac{a_{X0}x_{L}}{a_{nX0}}+\frac{b_{X0}-b_{nX0}}{a_{nX0}}=x_{L}^{*}$$


As the EqW is lower than the actual weight, this function must have a null intercept to retain its physical meaning. Therefore, the equivalence is

(26)
$$x_{L}^{*}=\sigma x_{L}=\frac{a_{X0}}{a_{nX0}}x_{L}$$


The slope (
$\frac{a_{X0}}{a_{nX0}}$
) is the coefficient that transforms the actual weight into the EqW, and 
$x_{L}^{*}$
 is the EqW.

## 3. Experimental Assessment

We devised experiments to illustrate the two proposed methods to compute the EqW. In these experiments, data were collected on the muscular activity of the erector spinae, with and without the assistance, provided by the exoskeleton. Trunk bending inclination and lifted weight were the independent variables of the system. The experiment required each subject to hold different weights in an upright and forward bending posture. The muscular activity was measured while the subject remained static in the required position for a certain time.

Both the increment of the load held and trunk bending require greater physical exertion of the trunk extensors to counteract the external forces and maintain balance [36,66,67]. This increased activation increases the risk of low back injuries due to the location of the trunk extensors with respect to the spine (as described earlier). Consequently, to study the overall effect of the exoskeleton on the trunk extensors, this analysis was conducted by measuring the muscular activation of both the erector spinae longissimus (ESL) and erector spinae iliocostalis (ESI). These activation levels were then averaged. This approach is used to consolidate the overall contribution of both muscles to a single net lumbar extensor activity: erector spinae (ES).

The experimental test was conducted with a sequence of semi-static postures and load configurations. To create the proposed scenario of static work with short periods of muscular rest, the index 
90th
 percentile was selected as it gives relevant information on the peak loads, with a probability level *p* = 0.9.

In addition to the activation of the extensor muscles, we also recorded the flexor muscles (rectus abdominis superior), to compute of co-activation indexes (CI). Variations in co-activation would provide important information on the effectiveness of the physical task and overall muscular activity, while simultaneously comparing the effect of the use of the exoskeleton.

### 3.1. Experimental Protocol

Twelve healthy adults participated in the study (six female, age 
42±3
 years, range: 39–49 years, height: 
1.66±0.06
 m, mass: 
59.3±8.7
 kg; six male, age: 
32±4
 years, range: 28–39 years, height: 
1.77±0.04
 m, mass: 
75.3±6.0
 kg). The experiment was approved by the Ethics Committee of Liguria (protocol reference number: CER Liguria 001/2019) and complies with the Helsinki Declaration. All the subjects signed a consent form prior to participating, after a full explanation of the experimental procedure. Subject selection was recommended if MMH was one of the subject’s job activities. Subjects participating in any clinical drug trials, subjects suffering from any neurological disorders and with orthopedic disorders possibly causing trunk and upper limb impairment, such as scoliosis, back pain, hand deformities, and painful musculoskeletal conditions, were excluded. No information regarding the expected results was provided to avoid bias, whether consciously or subconsciously.

Figure 3 shows how the experiments were conducted. Subjects were asked to hold four different loads: no load (N), 0 kg for both woman and men; low load (L), 4 kg for women and 5 kg for men; medium load (M), 8 kg for women, and 10 kg for the men; or high load (H), 12 kg for the women and 15 kg for the men. Two body postures were tested. In the first posture, the subjects stood in a neutral position (the upper arms in line with the torso and the forearms kept forward orthogonal to the upper arms). In the second posture, the subjects were bending forward, with an angle between the back and the coronal plane, of about 
$\theta =30^{\circ }$
. Each subject held the test loads in a static position for 10 s. Each subject performed all these tasks twice without the exoskeleton and twice wearing it, with conditions denoted, respectively, as noExo and ExoON. To avoid fatigue, there was at least two minutes of rest between each consecutive trial. The loads (N, L, M, and H) and the postures (upright or with a trunk bending of about 
$30^{\circ }$
) were assigned to each subject in a random order. Experimental tasks were performed first in the noExo condition and then in the ExoON condition. The exoskeleton was performing constant torque support for each experimental configuration as detailed in Section 2.1.

The loads were selected to not exceed the value of 1 with the NIOSH Lifting Index (LI). A LI value of 1.0 or less indicates a nominal risk to healthy employees. Therefore, the Recommended Weight Limit (RWL) for the tasks when not wearing the exoskeleton was calculated. The RWL is computed as described in (ISO TR 12295:2014). The upright position has all the RWL multipliers equal to one except for the Vertical Multiplier (VM), which is 0.93 (the mean vertical height of the hands above the floor is 
100cm±5cm
). Therefore, the RWL for the upright posture is calculated as: RWL = 0.93*LC (where LC is Load Constant, 25 kg for male and 20 kg for female). In the position with the trunk bent forward, VM is 0.96 (the mean vertical height of the hands above the floor is 
90cm±5cm
), while the Horizontal Multiplier (HM) is set to 0.63 as the horizontal distance is 
40cm±5cm
. The RWL for the forward bending posture is RWL = 0.60*LC. The NIOSH LI is computed using the reciprocal of the RWL as LI = L/RWL (where L is the given load).The two studied populations (female and male) have an LI of 0.33, 0.66, and 1 for the three lifted weights of L, M, and H, respectively.

Before measurements started, each subject underwent theoretical and practical training to learn the experimental procedure, including how to use the exoskeleton and become familiar with the device. An 8-channel Wi-Fi transmission surface electromyography (FreeEMG 300 System, BTS, Milan, Italy) was used to acquire the surface myoelectric signals (sEMG) at a sampling rate of 1000 Hz.

After skin preparation, bipolar Ag/AgCl surface electrodes (diameter 2 cm) prepared with electroconductive gel were placed over the muscle belly in the direction of the muscle fibers (distance of 2 cm between the center of the electrodes) according to the European recommendation for surface electromyography [68] and the atlas of muscle innervation zones [69]. The electrodes were placed bilaterally on two trunk extensors—the ESL and ESI—and on a trunk flexor, rectus abdominis superior (RAS). After the electrode placements, subjects performed (three times) a series of specific exercises [70,71] to record the isometric maximum voluntary contractions (MVC) for each of the investigated muscles. These exercises followed SENIAM recommendations [68].

### 3.2. Data Analysis

The data were processed using Matlab (MathWorks, Natick, MA, USA) software. sEMG signals were analyzed as in [36] following these three steps: (i) The MVCs and the sEMG raw data of each task were band-pass filtered using a fourth-order Butterworth filter (cut-off frequencies: 30–450 Hz); (ii) these filtered signals were then full-wave rectified and low-pass filtered using a fourth-order Butterworth filter at 5 Hz; and (iii) the rectified and filtered sEMG signals related to each task were normalized with respect to the maximum value of the corresponding muscle calculated as the mean of the maximum values detected for each of the three MVCs [68]. From the preprocessed sEMG signals of each task, to characterize differences in the muscle activation between different loads (N, L, M, H), trunk postures (upright vs. with a trunk bending of 
$30^{\circ }$
) and assistive conditions (noExo vs. ExoOn), we computed the 
90th
 percentile of the envelope of the preprocessed EMG signal [72,73,74].

Furthermore, we calculated the simultaneous activation of the trunk muscles (co-activation) using the time-varying multi-muscle co-activation function (TMCf) proposed in [66]:
(27)
$$TMCf\left(d\left(k\right),k\right)=\left(1-\frac{1}{1+e^{-12(d(k)-0.5)}}\right)\frac{{\left(\sum_{m=1}^{M}EMG_{m}\left(k\right)/M\right)}^{2}}{max_{m=1,..,M}\left[EMG\left(k\right)\right]}$$

where *d*(*k*) is the mean of the differences between each pair of 
$EMG_{m}\left(k\right)$
 at the 
kth
 sample, M is the number of considered muscles, and 
$EMG_{m}$
 is the signal of the 
mth
 muscle (ES and RAS). As co-activation index (CI) we considered the mean of the TMCf over each task.

We assessed the effect of the exoskeleton on the 
90th
 percentile and the CI through a two-way repeated measures ANOVA, taking into account the experimental conditions (noExo vs. ExoON) conducted at both 
$0^{\circ }$
 and 
$30^{\circ }$
 inclination, within each load (N, L, M, H) and their interaction. When a main effect of exoskeleton conditions or an interaction between the exoskeleton conditions and the load was found, post hoc analyses with Bonferroni’s correction were performed. The statistical analyses were conducted using SPSS 20.0 software (IBM). *p* values 
≤0.05
 were considered statistically significant.

### 3.3. Muscle Activation: Effect of Exoskeleton Assistance

#### 3.3.1. Upright: Trunk 
$0^{\circ }$
 Inclination

This section reports the results gathered with subjects standing in an upright position (
$0^{\circ }$
 trunk inclination), the two-way repeated measures ANOVA revealed a significant main effect from the wearing of the exoskeleton on the 
90th
 percentile index for the ES (measured as average muscular activation of the ESL and ESI with respect to the MVC of each of the two muscles). The effect of the interaction (exo*load) was detected on 
90th
 percentile for the ES (Table 1). Post hoc analysis showed significant differences between the noExo and ExoOn conditions at L (
p≤0.001
), M (
p=0.002
), and H (
p=0.002
) loads for the ES (shown in Figure 4). The condition N is not significant (
p=0.153
). For the co-activation index (CI) in this posture, the two-way repeated measures ANOVA showed a significant main effect from the exoskeleton assistance (Table 1). Post hoc analysis showed a significant difference on CI in the ExoON condition with respect to the noExo condition with all four loads (shown in Figure 4), 
p=0.023
 for the N load, 
p=0.006
 with the L and M loads, and 
p=0.001
 for the H load.

Considering the effect of the exoskeleton assistance, our results show that in the upright posture, the activation of the ES was reduced for all loads with respect to the standard configuration (without the exoskeleton), as shown in Figure 3. The activity reduction for the trunk extensors is due to the exoskeleton assistance, which contributes on the equivalence of both torque at the L5/S1 and compressing force shown in Equations (Equation 6) and (Equation 7). The assistance forces generated by the exoskeleton reduce the ES activity that normally would be generated; moreover, the forces generated by the exoskeleton are applied normal to spine thus do not contribute to the compression of the spine. Thus, the specific exoskeleton design [60] effectively reduces the muscular activity of the extensors [18,19]. The overall results on the whole test population, for the average muscular activity of the ES, show a reduction of about 32.5% across all the loads. This breaks down as a 34.1% reduction for the female population and a 30.1% reduction for males (results are shown in Table 2).

The co-activation index measured with the exoskeleton is always lower than without, except in the condition N (0 kg). This exception could be attributed to the unnatural assistance pattern/behavior of the exoskeleton (upright posture and no weight lifted) that increase the abdominal muscle contraction to counteract the force generated by the exoskeleton. As there is no ergonomic risk associated with this experimental configuration (N at 
$0^{\circ }$
), the exoskeleton should not provide any assistance (no ergonomic risk, therefore no assistance is required).

Figure 5 shows the presence of a linear trend in the muscular activation of both configurations (with and without the exoskeleton). The equation applied is 
$EMG_{\%}=a*x_{L}+b$
, where the 
EMG%
 is the average muscular activity generated for the specific load (
$x_{L}$
), the vector of coefficients 
[a,b]
 are the slope and intercept of the linear Equation (Equation 20) as presented in Section 2.3.2. The set of coefficients of both equations are 
$a_{n0}=0.72;b_{n0}=4.1;a_{x0}=0.4;b_{x0}=3.29,$
 where the index “n0” is for the condition noExo at 
$0^{\circ }$
 inclination, while “x0” is for the ExoON at 
$0^{\circ }$
 inclination. The goodness of the interpolation has been evaluated using the 
$R^{2}$
 and RMSE factors and reported in Table 5 for both conditions and populations, at 
0∘
 inclination, noExo and ExoON. The fitting result, obtained considering the 
$R^{2}$
 index, indicates a worst-case fitting of at least 98% on both populations and an average RMSE of 0.23% of MVC.

#### 3.3.2. Trunk 
$30^{\circ }$
 Inclination

When the subjects were at a 
$30^{\circ }$
 forward trunk inclination, the two-way repeated measures ANOVA revealed a significant main effect when using the exoskeleton on 
90th
 percentile for the ES (Table 3). As regards the ES, post hoc analysis showed a significant decrease in the ExoOn condition with respect to the noExo condition on 
90th
 percentile for all the four loads as shown in Figure 6, 
p≤0.001
 with the N and L loads, 
p=0.001
 for the M load and 
p=0.004
 for the H load. For the CI, in this posture the two-way repeated measures ANOVA showed a significant main effect from the exoskeleton assistance and of the interaction with loads (EXO*Load) (Table 3). Post hoc analyses showed a significant decrease in the CI in the ExoOn condition with respect to the noExo condition at load configurations N (
p=0.001
), L (
p=0.001
), and H (
p=0.001
) (shown in Figure 6). The condition M is not significant (
p=0.053
).

In the posture with 
$30^{\circ }$
 trunk bending, the results on the 
90th
 percentile are consistent with the those found in the upright posture, but are more pronounced; in fact, there is a significant reduction in muscle activity for both the ES and CI indexes, thus showing greater efficacy of the exoskeleton at this inclination. The overall results on the ES show an average reduction in muscular activity of about 
44.25%
 across the whole population and for all four loads (shown in Table 4). A higher reduction is observed in the female population reaching an average of 
45.45%
, while males show a reduction of 
42.95%
.

This more pronounced effect of the exoskeleton in the posture with 
$30^{\circ }$
 trunk bending with respect to the upright posture could be because, in this more physically demanding condition, the exoskeleton is more effective.

This experimental configuration shows a linear trend in the muscular activation for both configurations with and without the exoskeleton (shown in Figure 7). The linear equation is 
$EMG_{\%}=a*x_{L}+b$
, where the 
EMG%
 is the average muscular activity generated for the specific load (
$x_{L}$
), and the coefficients 
[a,b]
 are the slope and intercept of the linear equation, respectively. The set of coefficients of both equations is 
$a_{n30}=0.71$
; 
$b_{n30}=11.5;a_{x30}=0.41;b_{x30}=6.5$
. The goodness of the interpolation has been evaluated using the 
$R^{2}$
 and RMSE factors and reported in Table 5 for both conditions and populations, at 
0∘
 inclination, noExo and ExoON. The fitting result, obtained considering the 
$R^{2}$
 index, indicates a worst-case fitting of at least 97% on both populations and an average RMSE of 0.53% of MVC.

#### 3.3.3. Equivalent Weight Results

The EqW is computed based on both approaches proposed earlier. The first method uses Equation (Equation 19) in Section 2.3.1, where the equation inputs are the normalized muscular activity of the whole ES group in the standard condition (noExo), the muscular reduction due to wearing the exoskeleton, the lifted weight, eventual body inclination, and the anthropometric measurements of each subject (whole body weight and height). The second method applies the computation presented in Section 2.3.2 (Equation (Equation 26)), and it takes as its system inputs the muscular activations of the ES group in both configurations, with and without the exoskeleton.

The application of Equation (Equation 19) (more details are reported in the Appendix A) allows the estimation of the coefficient 
ψ
, which defines the apparent reduction in the lifted weight. The 
ψ
 values obtained for the test configurations and population are reported in Table 6. The coefficient 
ψ
 is normalized on the angle component due to the body inclination, thus 
ψ
 does not vary while considering the two different body inclination (
$0^{\circ }$
, and 
$30^{\circ }$
). The overall weight reduction found was 
ψ=0.67
. This means that, when assisted by this exoskeleton, the user perceives a weight over a third lighter than the actual loading. For example, a 15 kg load is perceived as being about 10.05 kg, which is the termed EqW.

Figure 8 shows the results obtained by applying the second method. Figure 8A shows the trends for the overall population and female and male populations at 
$0^{\circ }$
 inclination, while Figure 8B shows the three trends at 
$30^{\circ }$
 inclination. The coefficients 
σ
 of the EqW are reported in Table 7. The results show an apparent reduction in the lifted weight (EqW) of about 
σ=0.58
 for the overall population. By applying this coefficient, a 15 kg load appears as an EqW of 8.7 kg.

Finally, we note that the investigated coefficients 
ψ
 and 
σ
 confirm the assessment of the EqW, returning similar values for the EqW, which is just below of the two third of the actual lifted weight.

## 4. Discussion

Low Back Pain (LBP) is one of the major occupational health concerns [7,75], but despite the extensive research and mitigation actions, the causes of LBP are still elusive and treatment effects are unsatisfactory [76,77]. Companies invest substantial resources in their occupational health and safety professionals and use ergonomic assessments to mitigate the risk of injury when workers carry out manual material handling tasks. To reduce some known or suspected risk factors, current practice is based on the NIOSH guidelines. The NIOSH Lifting Index (
LI
) considers several job task variables to assess the risk associated with the task. The computation of the LI is based on the Recommended Weight Limit (RWL), which defines the maximum acceptable load that employees can safely lift over the course of an 8-hour shift without increasing the risk of MSD to the lower back.

Recently, collaborative exoskeletons have been developed and introduced into the market to help workers reduce their exposure to key risk factors. There has been encouraging evidence of their potential [25] to mitigate risk. However, an important barrier to the large-scale adoption of exoskeletons remains: the impossibility to quantify the ergonomic risk reduction. The rationale behind this study is to search common ground between exoskeleton developers, Health and Safety professionals, and other stakeholders to foster discussion and overcome this barrier. To this end, we have characterized the effects of assistance generated by an active back-support exoskeleton in terms of apparent lifted weight reduction. Results are supported by a theoretical analysis using the biomechanical model detailed in Section 2.2, and these clearly show the relationship between lower back muscular reduction and the apparent decrease of lifted weight.

In this study, we assessed the reduction of the trunk muscles activation, through EMG [78], while supporting four different loads in a static posture, with the trunk vertically upright and with a forward inclination of 
$30^{\circ }$
. These tests were conducted both with and without the assistance of the XoTrunk exoskeleton. To allow the results of this study to be used in an eventual ergonomic risk assessment method, we apply the reduction of the lifted weight by multiplying it by the EqW coefficient (
ψ
 or 
σ
), thus taking into consideration the assistance of the exoskeleton. Thus, the EqW determines an effective reduction of the lifted weight according to 
$L^{*}=\sigma L\left(or,=\psi L\right)$
, where *L* is the effective lifted load and 
$L^{*}$
 is the apparent lifted load. Consequently, in the conducted study the three lifted weights (L, M, and H) would be apparently considered as 
$L^{*}=0.625L=\left[3.12,6.25,9.38\right]$
 kg, where the lifted weights L are 
[5,10,15]
 kg and 0.625 is the average value of 
σ
 and 
ψ
.

It is worth mentioning that the CI is mainly used as a diagnostic marker to assess the appropriateness of the assistance generated by the exoskeleton from the user’s perspective. It is evident that the exoskeleton does not create any mismatch between the CI and the other monitored indexes showing a reduction of muscular activity. The only condition in which a different pattern emerged is the condition N (0 kg, no external load) at 
$0^{\circ }$
 trunk inclination. This is attributable to the unnatural assistance of the exoskeleton in this condition (upright posture and no weight lifted), which causes an increase in abdominal muscles contraction to counteract the force generated by the exoskeleton. Given the absence of ergonomic risk associated with this experimental configuration, the exoskeleton should not be providing any assistance.

## 5. Conclusions

The study presented in this paper investigates a possible approach to define a relationship between the use of assistive lower back exoskeletons and the reduction of loading at the back. This will mitigate the ergonomic risk associated with MMH activities and establish a common ground that provides a means of reliable assessment and discussion between exoskeletons designer/evaluators and Health and Safety professionals. Given extensive experimental observations on the muscular activities of a subset of tasks, we give a method to approximate a relationship between the weight being held and the weight that the spine “experiences” when assisted by the device. This relationship, which we called EqW, consists of an apparent reduction of the actual lifted weight due to the effective lower back muscular activity reduction. This reduction was estimated using two different methods and applied to the exoskeleton prototype XoTrunk. The two methods led to very similar numerical results: 
σ
 is 0.58 and 
ψ
 is 0.67 giving an average of 0.625, meaning that a lifted weight of 15 kg appears as only 9.38 kg to the exoskeleton wearer. The EqW coefficient was proposed to be introduced as part of ergonomic assessment procedures. The study proposes its use in place of the actually lifted weight for future ergonomic assessment methods that may take into consideration the assistive effect of exoskeletons for a given activity. Moreover, the two proposed methods can be also employed a posteriori on datasets of exoskeleton performance evaluation. Note that as the study is only validated up to 15 kg of weight lifted and the same performance as EqW may not be generalized directly to a higher range of loads, so future studies will aim to evaluate the same approach covering lifting loads to the limit of the permitted weight.

These results have been gathered and validated through an experimental campaign involving 12 subjects (six male and six female), four different loads, and two different postures. This study was also designed to assess the relevance of substantial variability across subjects holding different weights and postures to understand whether the benefit of providing assistance was affected or varied, resulting in a low variability of results even across the male and female populations.

The CI, used as a diagnostic marker, assesses the appropriateness of assistance generated by the exoskeleton in all the studied conditions apart from the condition N at 
$0^{\circ }$
. For this condition, there is no ergonomic risk, thus no assistance should be provided.

This method is in principle applicable to all types of back-support exoskeletons, but distinctions would be necessary between active/passive, rigid/soft, or parallel/perpendicular devices as their assistive patterns/behaviors can be substantially different. Furthermore, this method is valid if the exoskeleton does not alter the user’s movements and dynamics.

The present study considers two static holding tasks, which are a subset of MMH activities; thus, future development will be to extend this analysis to a variety of tasks. Moreover, the exoskeleton was assisting the user by applying a constant torque during the experiment. This control strategy was adopted due to the static nature of the experimental section, which might not be always the case in a more dynamic activity [56]; thus, future works will consider the application of this approach on more complex working scenarios.

Even though this study is based on well-founded theory, it is vital that this method should be validated using an epidemiological approach to investigate the association between Lifting Index values and health outcomes; also, studies on specific worker’s age populations and counterbalancing the order of experimental conditions could be considered.

## Figures and Tables

**Figure 1 ijerph-18-02677-f001:**
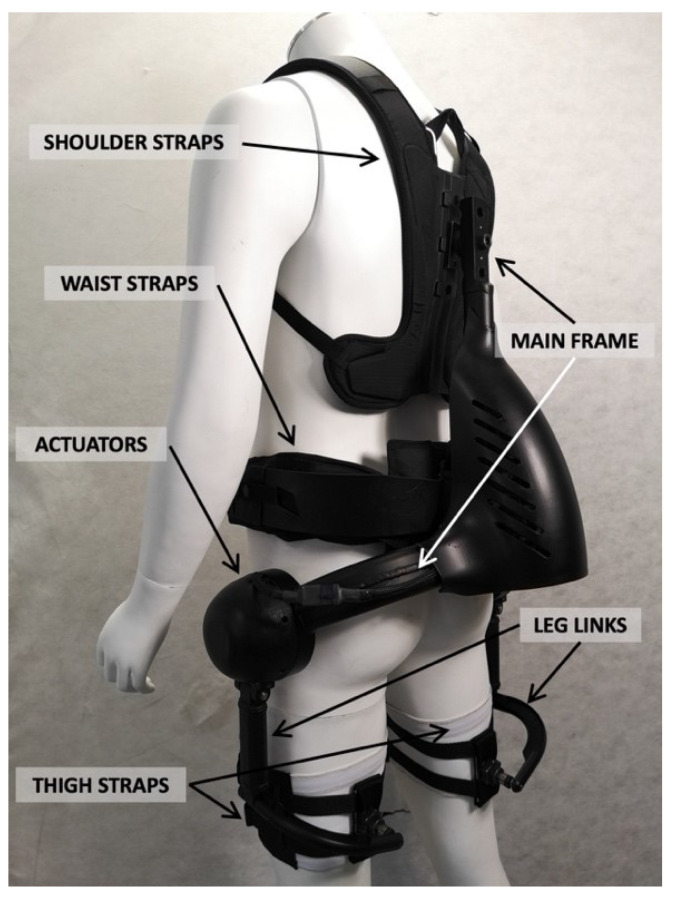
XoTrunk prototype.

**Figure 2 ijerph-18-02677-f002:**
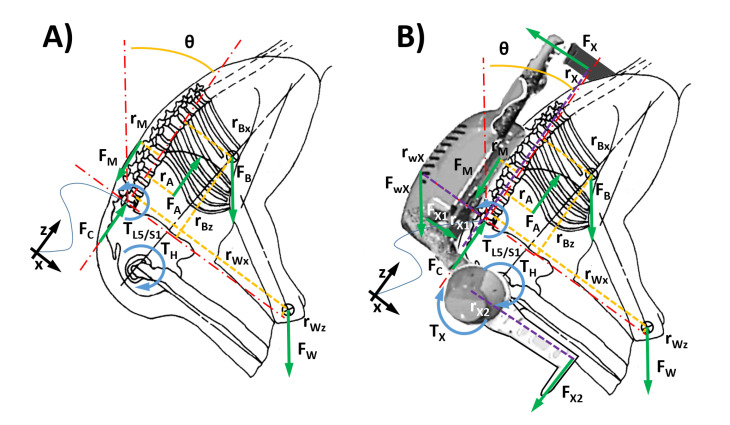
(**A**) Biomechanical model showing forces and moments operating on the L5/S1 joint during load lifting. (**B**) Biomechanical model showing forces and moments operating on the L5/S1 joint during load lifting when assisted by a back-support exoskeleton.

**Figure 3 ijerph-18-02677-f003:**
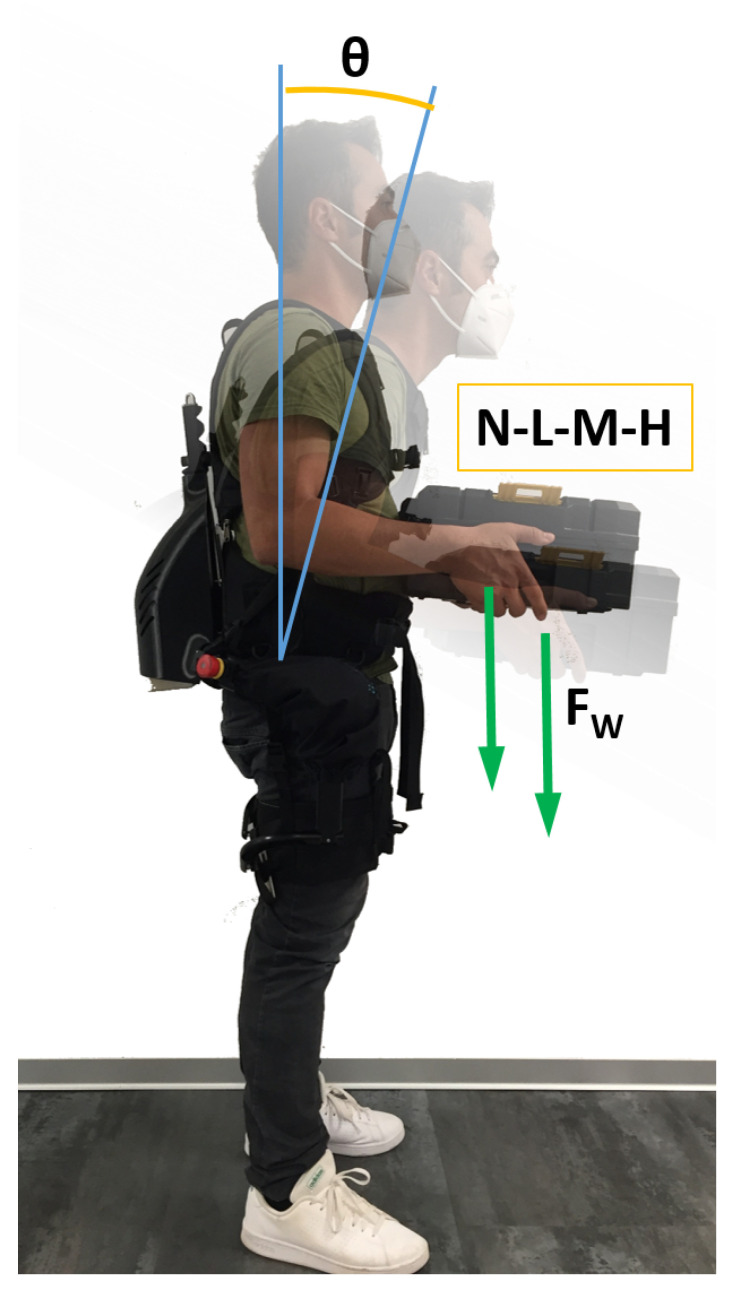
Experimental protocol overview.

**Figure 4 ijerph-18-02677-f004:**
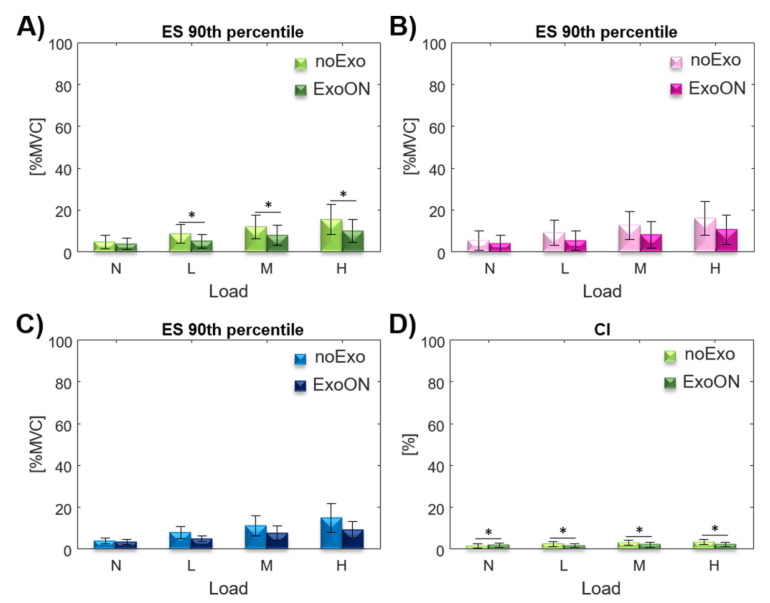
90th
 percentile of the ES muscular activation, with and without the exoskeleton, at 
$0^{\circ }$
 inclination and four load conditions for the whole population (**A**), for female (**B**), and male (**C**) populations. (**D**) CI, with and without the exoskeleton, in the four load conditions.

**Figure 5 ijerph-18-02677-f005:**
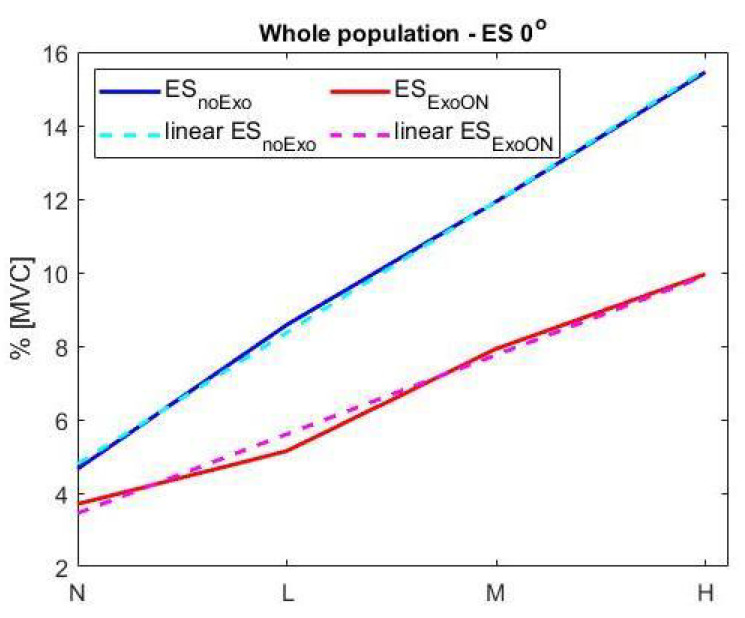
Interpolation trend of 
90th
 percentile of both ES muscular activation, with and without the exoskeleton, at 
$0^{\circ }$
 inclination for the four load conditions.

**Figure 6 ijerph-18-02677-f006:**
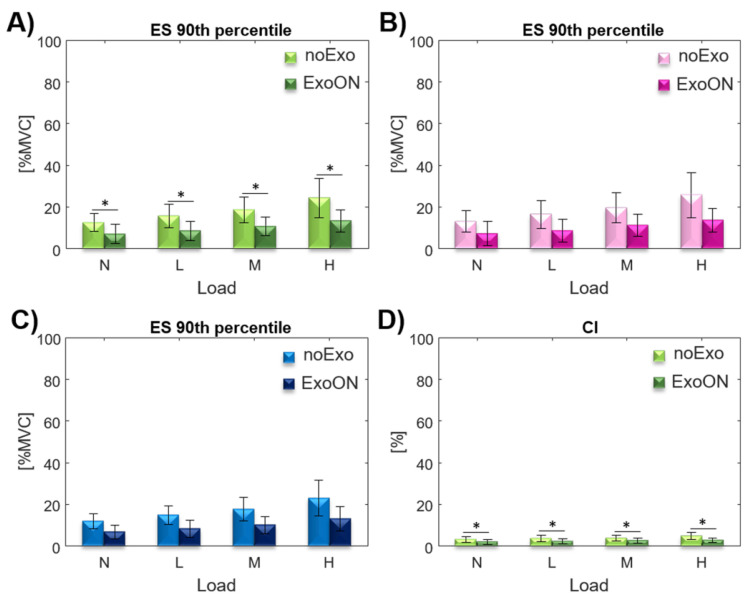
90th
 percentile for the ES muscular activation, with and without the exoskeleton, at 
$30^{\circ }$
 inclination and for four load conditions for the whole population (**A**), the female population (**B**), and the male (**C**) population. (**D**) CI with and without the exoskeleton in the four load conditions.

**Figure 7 ijerph-18-02677-f007:**
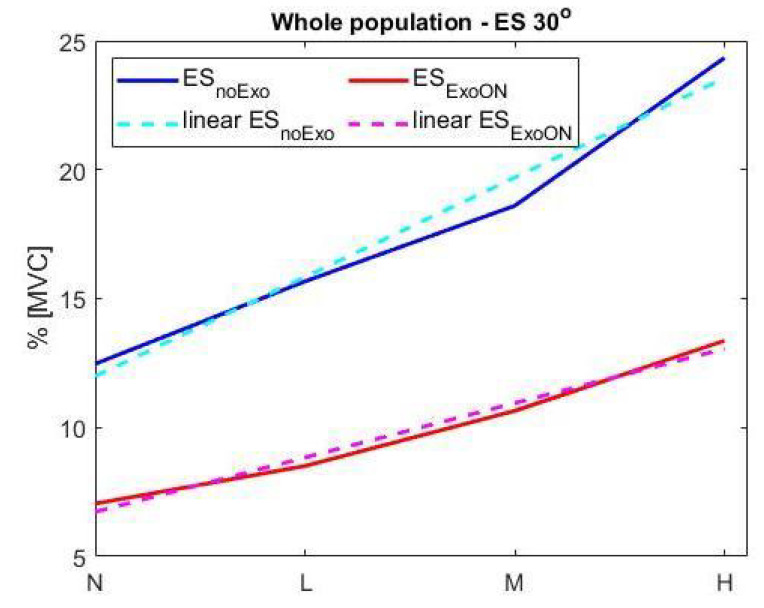
Interpolation trend of 
90th
 percentile of both ES muscular activation with and without the exoskeleton at 
$30^{\circ }$
 inclination for the four load conditions.

**Figure 8 ijerph-18-02677-f008:**
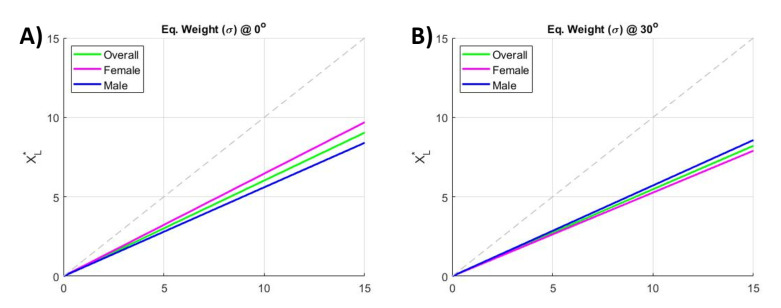
(**A**) EqW (
σ
) trend for the whole population and female and male populations at 
$0^{\circ }$
 inclination for the four load conditions. (**B**) EqW trend for the whole population and female and male populations at 
$30^{\circ }$
 inclination for the four load conditions.

**Table 1 ijerph-18-02677-t001:** Summary of statistical analysis on exoskeleton effect for the 
$0^{\circ }$
 trunk inclination.

Index	Main Effect-EXO		Main Effect-EXO*Load	
90th ES	F(1,11) = 22.384	p=0.001	F(1.291,14.202) = 14.855	p≤0.001
CI	F(1,11) = 11.769	p=0.006	F(1.345,18.805) = 14.795	p=0.154

**Table 2 ijerph-18-02677-t002:** Summary of the results and descriptive statistics (mean ± STD) due to the exoskeleton effect for the 
$0^{\circ }$
 trunk inclination. (†) the unit is [%MVC], RR stands for Reduction Rate, and TR stands for Trend.

Category	Index	Modality	N ${}^{†}$	L ${}^{†}$	M ${}^{†}$	H ${}^{†}$	TR
Overall	90th ES	noExo	4.66 ± 3.19	8.58 ± 4.30	11.93 ± 5.37	15.45 ± 6.77	3.32
Overall	90th ES	ExoON	3.7 ± 2.66	5.14 ± 3.09	7.92 ± 4.64	9.95 ± 5.18	2.69
Overall	90th ES	RR	20.6%	40.1%	33.6%	35.6%	32.5%
Male	90th ES	noExo	3.93 ± 1.26	8.04 ± 2.63	11.16 ± 4.38	14.93 ± 6.20	3.8
Male	90th ES	ExoON	3.43 ± 1.10	4.95 ± 1.20	7.65 ± 3.09	9.27 ± 3.45	2.7
Male	90th ES	RR	12.7%	38.4%	31.5%	37.9%	30.1%
Female	90th ES	noExo	5.38 ± 4.20	9.11 ± 5.44	12.7 ± 6.11	15.98 ± 7.25	2.97
Female	90th ES	ExoON	3.97 ± 3.58	5.33 ± 4.20	8.2 ± 5.78	10.63 ± 6.39	2.68
Female	90th ES	RR	26.2%	41.5%	35.4%	33.5%	34.1%
Overall	CI	noExo	1.45 ± 1.03	2.46 ± 1.21	3.05 ± 1.29	3.38 ± 1.16	2.33
Overall	CI	ExoON	2.02 ± 0.97	1.67 ± 0.85	2.25 ± 1.13	2.25 ± 0.90	1.11

**Table 3 ijerph-18-02677-t003:** Summary of the statistical analysis of the exoskeleton effect for the 
$30^{\circ }$
 trunk inclination.

Index	Main Effect-EXO		Main Effect-EXO*Load	
90th ES	F(1,11) = 22.657	p=0.001	F(1.102,12.122) = 4.479	p=0.053
CI	F(1,11) = 19.355	p=0.001	F(1.833,20.165) = 10.209	p=0.001

**Table 4 ijerph-18-02677-t004:** Summary of the results and descriptive statistics (mean ± STD) due to the exoskeleton effect for the 
$30^{\circ }$
 trunk inclination. (†) the unit is [%MVC], RR stands for Reduction Rate, and TR stands for Trend.

Category	Index	Modality	N ${}^{†}$	L ${}^{†}$	M ${}^{†}$	H ${}^{†}$	TR
Overall	90th ES	noExo	12.47 ± 4.12	15.67 ± 5.30	18.6 ± 6.04	24.34 ± 9.03	1.95
Overall	90th ES	ExoON	7.05 ± 4.28	8.51 ± 4.38	10.65 ± 4.30	13.37 ± 5.14	1.89
Overall	90th ES	RR	43.5%	45.7%	42.7%	45.1%	44.3%
Male	90th ES	noExo	11.94 ± 3.30	14.88 ± 4.18	17.67 ± 5.25	22.91 ± 7.84	1.92
Male	90th ES	ExoON	6.86 ± 3.00	8.39 ± 3.72	10.1 ± 3.74	13.09 ± 5.22	1.91
Male	90th ES	RR	42.5%	43.6%	42.8%	42.9%	42.9%
Female	90th ES	noExo	13.0 ± 4.75	16.45 ± 6.12	19.54 ± 6.60	25.77 ± 9.88	1.98
Female	90th ES	ExoON	7.23 ± 5.25	8.62 ± 4.95	11.19 ± 5.74	13.64 ± 5.05	1.89
Female	90th ES	RR	44.4%	47.6%	42.7%	47.1%	45.5%
Overall	CI	noExo	3.13 ± 1.32	3.62 ± 1.42	3.69 ± 1.30	4.75 ± 1.61	1.52
Overall	CI	ExoON	1.98 ± 1.09	2.26 ± 1.17	2.6 ± 1.18	2.76 ± 1.05	1.4

**Table 5 ijerph-18-02677-t005:** RMSE and 
$R^{2}$
 values of the linear interpolations.

Condition	Population	$R^{2}$	RMSE
noExo@ $0^{\circ }$	Female	99.92%	0.115
ExoON@ $0^{\circ }$	Female	98.22%	0.344
noExo@ $30^{\circ }$	Female	97.17%	0.797
ExoON@ $30^{\circ }$	Female	98.49%	0.302
noExo@ $0^{\circ }$	Male	99.75%	0.202
ExoON@ $0^{\circ }$	Male	98.77%	0.252
noExo@ $30^{\circ }$	Male	97.44%	0.646
ExoON@ $30^{\circ }$	Male	97.21%	0.386

**Table 6 ijerph-18-02677-t006:** Overall coefficient 
ψ
 for different weights, trunk inclinations, and gender populations.

Population	ψ @ $0^{\circ }$	ψ @ $30^{\circ }$	ψ Mean
Overall	0.67	0.67	0.67
Male	0.59	0.69	0.64
Female	0.73	0.66	0.695

**Table 7 ijerph-18-02677-t007:** Overall coefficient 
σ
 for different weights, trunk inclinations, and gender populations.

Population	σ @ $0^{\circ }$	σ @ $30^{\circ }$	σ Mean
Overall	0.61	0.55	0.58
Male	0.56	0.57	0.565
Female	0.65	0.53	0.59

## Data Availability

The data presented in this study are available on request from the corresponding author.

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
