# Peer review of "Equivalent Weight: Connecting Exoskeleton Effectiveness with Ergonomic Risk during Manual Material Handling"

_ijerph, 2021, doi:10.3390/ijerph18052677_

Round 1

Reviewer 1 Report

This manuscript mainly proposes the use of Equivalent weight to evaluate the waist assisted exoskeleton, and introduces the CI index to assist the transporter to evaluate the risk. It is a very interesting manuscript to evaluate the effectiveness of the wearable waist assisted exoskeleton, but the author needs to solve the following problems :

  1. Regarding the Xo-Trunk exoskeleton, the author only briefly introduced the mechanical structure,  A short brief needs to be added  how the system achieve motion control and waist assistance.

2.In the process of the exoskeleton assisted wearer experiment, both men and women participated in the experiment, but obviously the ages of men and women are concentrated in a range, and the age distribution of the subjects should be more even

3.Regarding the exoskeleton assisting human body transportation, why the author measured the EMG signal of the erector spinae muscles of the back during static maintenance (0° or 30°)? due to the manual material handling process should be a dynamic process. the comparison of the entire back EMG signal during the dynamic process should also be discussed. Moreover, because the system outputs a constant torque regardless of the posture, this is also Rather strange,The author should compare the electromyographic signal when the motor output torque of the hip joint is planned.

  1. The angle range of the experiment is too scarce. Only 30° and 0°

Author Response

Reviewer Comment 1.1 (RC1.1): Regarding the Xo-Trunk exoskeleton, the author only briefly introduced the mechanical structure,  A short brief needs to be added  how the system achieve motion control and waist assistance.

Authors Answer (AA): The authors thank the reviewer, and we add a short description of the exoskeleton control in section 2.1. The change in the manuscript is as follows.

RC1.2: In the process of the exoskeleton assisted wearer experiment, both men and women participated in the experiment, but obviously the ages of men and women are concentrated in a range, and the age distribution of the subjects should be more even

AA: The authors thank the reviewer for the comment, these are characteristics of the studied population: six female, age 42 ± 3 years, range 39-49 years, height 1.66 ± 0.06 m, weight 59.3 ± 8.7 kg; six male, age 32 ± 4 years, range 28-39 years, height 1.77 ± 0.04 m, weight 75.3 ± 6.0 kg. Workers who have manual material handling as part of their job activities and physically fit for the job were selected, and these were the available populations. In a future extended study, the age of the population will be taken in consideration and more even age will be chosen. This is stated in the future work of conclusion as follows: […] Even though this study is based on well-founded theory, it is vital that this method should be validated using an epidemiological approach to investigate the association between Lifting Index values and health outcomes, also studies on specific worker’s age populations could be considered.  

RC1.3: Regarding the exoskeleton assisting human body transportation, why the author measured the EMG signal of the erector spinae muscles of the back during static maintenance (0° or 30°)? due to the manual material handling process should be a dynamic process. the comparison of the entire back EMG signal during the dynamic process should also be discussed. Moreover, because the system outputs a constant torque regardless of the posture, this is also Rather strange,The author should compare the electromyographic signal when the motor output torque of the hip joint is planned.

AA: The authors thank the reviewer for the comment. The authors have completed many dynamic studies by measuring the muscle activity of the erector spinae, but for this study we chose to start from static tasks to simplify the interpretation of the results. The simplification of the protocol is sometime important to underline the main effect of the biomechanical system useful for studying the main contribution of a static biomechanical model. A dynamic test would require additional factors to be considered into the equations, introducing greater uncertainty that would reduce the reliability of the results and at this stage raise more questions than provide answers. The reason why the authors decide to design a static test is to consolidate this step before move on building on a more complete and difficult scenario where also dynamics will be taken in consideration. Therefore, in the future work the necessity of conducting a dynamic test to validate this approach further is stated as follows: […] Moreover, the exoskeleton was assisting the user by applying a constant torque during the experiment. This control strategy was adopted due to the static nature of the experimental section, which might be not always the case in a more dynamic activity [45], thus, future works will consider the application of this approach on more complex working scenarios.[...]. Concerning the constant torque, it is quite common that exoskeletons provide torque assistive profile which is usually function of the trunk inclination or proportional to other sensory signals, therefore, if these signals are constant (e.g. trunk inclination angle) also the torque output is constant as well, therefore in a static test there is high probability to receive a constant signal even if the controller is not programmed to generate a constant signal as in our scenario.    

RC1.4: The angle range of the experiment is too scarce. Only 30° and 0°

AA: The authors thank the reviewer for the comment. The angles were selected as representative of two typical inclination covered by workers that hold incongruous postures (30 deg) or accomplish lifting tasks where 30 deg and 0 deg are the initial and final positions of the task. It might be also more but considering the lifted weight the NIOSH lifting index rise up to 1, thus the subject is approaching the recommended limits for the safety of the activity. Therefore, we preferred to make the subject perform in a safe condition not to exaggerate with a more extreme posture.

Reviewer 2 Report

The authors have attempted to quantify the effectiveness of a back support exoskeleton by introducing a coefficient which depicts how much of weight reduction the person experiences while holding loads statically compared to the actual weight of the load. This approach can be of value and interest to several people in the exoskeleton industry or health and safety professionals. However, I have some concerns regarding this manuscript.

One of my major concerns is that, in the Exoskeleton prototype section you have stated a constant amount of torque was generated by the exoskeleton throughout the experimental task. Did this torque remain the same for both postures? Or did the exoskeleton adjust the torques based on the posture and weight of the load? Please consider including further details for clarification.

Another major concern that I have is that in the Biomechanics of load handling section, you have stated that the torque generated by the exoskeleton causes generation of force in the waist belt, shoulder straps and thigh braces contact points. However, only the force generated at the waist belt has been included in the remainder of the equations.  There is concern that while low back muscle activity may decrease hip and shoulder muscle activity may increase. Lack of data for those muscle groups creates an issue for properly interpreting the results.

Another major concern is that in the biomechanical model, the added torque on the L5/S1 joint due to the weight of the exoskeleton is not taken into consideration for the conditions in which the participant is wearing the exoskeleton.

Another major concern that I have is regarding the data interpolation model. How are we certain that there is a linear relationship between the weight of the lifted load and the normalized EMG value?

I also have some minor concerns that are as follows:

Introduction- line 53, change workers to worker.

Introduction- line 55-66, Please consider including more details regarding the results of the studies you have referenced for the 5 assessment techniques? What are some results of these assessments in the literature? This can help your argument of exoskeletons being effective.

Introduction- line 78-79, you have claimed a poor correlation between direct measurements of physiological parameters and observational techniques such as NIOSH lifting index, OCRA, etc. Please consider including further details and citations for this claim.

Introduction- line 94-98, depending on how much space or page limit allows, please consider making a stronger argument about the current barriers for the common use of exoskeletons and/or list more citations.

Materials and methods- line 151, change a relatively small moment arms to relatively small moment arms

Materials and methods- line 183- 184, change the the abdominal force to the abdominal force

Materials and methods- Figure 2B: please consider including the rx, rx1 and rx2 moment arms in the figure for better visualization.

Materials and methods- line 196-197, Is rx2 missing from this sentence? Please clarify what rx2 refers to.

Materials and methods- Equations 4, please provide explanation for why Fx and Fx1 are equal. Is there a reason the torque due to the weight of the exoskeleton is not taken into account here?

Materials and methods- line 224, at the end of the line add a space between defined and between

Materials and methods- line 227-228, did you use body weight or body mass for this equation? The text states body weight but the unit in the equation states kg which is a unit of mass. Please change one of them and make them consistent with each other.

Materials and methods- line 228, please include more details about this equation. How was it developed? Please include details or more clear citations. Is it taken from reference 52?

Materials and methods- line 229, The torques due to abdominal force and the load are not included in this equation. Why? Please include some explanation regarding this equation or provide a clearer citation to help the reader.

Materials and methods- line 311-312, the intercept can have a negative value and still have a physical meaning. Please include some explanation to why the intercept cannot be negative.

Materials and methods- line 321, change remains to remained.

Experimental assessment- line 333-336, what does this paragraph mean? The 90th percentile of what?

Experimental assessment, line 343 and 344- change weight to mass.

Experimental assessment- line 353, change unconsciously to subconsciously.

Experimental assessment- line 367, not counterbalancing the order of experimental conditions (with or without the exoskeleton) can potentially affect the results. Please consider stating this as one of the limitations of the study since the results might be affected.

Experimental assessment- line 391, on which side of the body were the EMG electrodes placed? Right or left? And why? Please specify

Experimental assessment- line 404-407, similar to one of the previous comments, what does this mean? 90th percentile of what?

Experimental assessment- line 416-417, what about the effect of the inclination? Please explain why the main effect of the inclination and its interaction with the other two factors was not included in the statistical model.

Experimental assessment- line 430, is there an L or N missing before the first reported p value representing the low load or the no load condition? Please include all four p- values even if there were not statistically significant.

Experimental assessment- line 438, change load to loads

Experimental assessment- Table 2, what is the tendency rate? Please include a description

Experimental assessment- line 474, what about the p value for the M condition? Please report the value even if it is not significant.

Experimental assessment- Table 5 and Table 6, why do you think the coefficient for the weight reduction is a larger number for females in both techniques? Please include relevant citations if needed.

Experimental assessment- line 515, change confirms to confirm.

Discussion- line 532-534, please consider rephrasing this sentence. There are viable methods to quantify the effectiveness of exoskeletons, they may just not be simple enough, but lack of viable methods is not accurate.

Discussion- line 547, change multiply to multiplying.

Discussion- line 551, change consider to considered

Discussion- line 554, please consider removing the word index. CI stands for coactivation index. Therefore, CI index is redundant. There are further examples of this redundancy in this section.

Correlation- line 566, please consider replacing the word correlation with a more suitable word. Mentioning correlation requires performing a correlation statistical analysis. Therefore, use a more accurate wording.

Conclusion- line 600- change might be not to might not be

Appendix A- line635 and 641, what are these question marks for?

Appendix A- equation A2, change trunck to trunk. There are further examples of this misspelling throughout the appendix.

Author Response

RC2.1: The authors have attempted to quantify the effectiveness of a back support exoskeleton by introducing a coefficient which depicts how much of weight reduction the person experiences while holding loads statically compared to the actual weight of the load. This approach can be of value and interest to several people in the exoskeleton industry or health and safety professionals. However, I have some concerns regarding this manuscript.

AA: The authors thank the reviewer for the comment.

RC2.2: One of my major concerns is that, in the Exoskeleton prototype section you have stated a constant amount of torque was generated by the exoskeleton throughout the experimental task. Did this torque remain the same for both postures? Or did the exoskeleton adjust the torques based on the posture and weight of the load? Please consider including further details for clarification.

AA: The authors thank the reviewer for the comment. The torque generated was constant for all postures. While it is generally desirable for the exoskeleton to adjust its torque based on posture, we made the choice of constant torque to simplify the study design and this make it easier to interpret the results. Torques were set with two different values for the two female and male populations, with female receiving 80% of the male torque value, this was chosen to also respect the difference in the lifted weight between women and men (also with a proportion of the 80% on the lifted weight). This is due to the maximum limit of weight that men and women has while applying the standard of the manual material handling. The consideration on the torque applied has been clarified in the manuscript as follows: section 2.1 […] In the present study, the control algorithm was simplified to fit the experimental protocol and the static nature of the physical tasks performed (more details in Section 3.1). Specifically, the exoskeleton was programmed to generate constant torques (24Nm for women and 30Nm for men) during the experimental task. For male participants we selected 30Nm torques, applying the authors’ previous experience [45] to the static nature of the assessed task. For female participants, we used the same proportion found in the reference masses suggested by the ISO TR 12295:2014 (25kg for men and 20kg for women), and therefore selected 24Nm constant torque for women. The main reason to constrain the system to provide a constant value of torque during the experimental section was mainly for safety reasons, preventing form eventual oscillations of the torque reference due to possible system noise. Section 3.1 […] Experimental tasks were performed first in the noExo condition and then in the ExoON condition. The exoskeleton was performing constant torque support for each experimental configuration as detailed in Section 2.1. […]

RC2.3: Another major concern that I have is that in the Biomechanics of load handling section, you have stated that the torque generated by the exoskeleton causes generation of force in the waist belt, shoulder straps and thigh braces contact points. However, only the force generated at the waist belt has been included in the remainder of the equations.  There is concern that while low back muscle activity may decrease hip and shoulder muscle activity may increase. Lack of data for those muscle groups creates an issue for properly interpreting the results.

AA: The authors thank the reviewer for the comment. The only force that act on the reduction of the lower back muscles is the one generated by the exoskeleton at the shoulders; the force at the waist is the reaction force and pivot for the exoskeleton. The forces generated at the legs are also helping while a dynamic lifting motion is carried out, thus the force at the legs help the wearer to lift up a weight by assisting the hips. This effect (the force at the legs) does not affect or benefit in any way the lower back muscles, which are the main responsible to musculoskeletal disorder due to manual material handlings. In this particular experimental protocol, in which the experimenter asked explicitly to the subject to hold the position for few seconds, the muscles of the legs might increase their muscle activities to counter the effect generated by the exoskeleton on the hips. In addition, the shoulder are not influenced by the application of the forces generated by the exoskeleton. This force is the one that pulls back the subject countering the weight lifted. Therefore, the authors consider the analysis sufficient for the specific test, while future studies could include the observation of the effect on the hips for a more complete picture.

RC2.4: Another major concern is that in the biomechanical model, the added torque on the L5/S1 joint due to the weight of the exoskeleton is not taken into consideration for the conditions in which the participant is wearing the exoskeleton.

AA: The authors thank the reviewer for the comment. The weight of the exoskeleton is load on the waist of the user. The exoskeleton design has been specifically developed to load its weight on the user waist, which is below the L5/S1 thus no torque is applied to the spine due to the exoskeleton weight. But also, even if the total weight of the device was loaded on the upper body, 6kg would translate to approximately 60N compression, which is arguably negligible compared to the recommended limit of 3400N. This is better clarified in the manuscript. Section 2.1: […] Details on the kinematics and physical attachments on XoTrunk are reported in [43]. The waist belt, is designed to secure most of the weight of the exoskeleton (7.5 kg) on the hips over the iliac crest [44].

RC2.5: Another major concern that I have is regarding the data interpolation model. How are we certain that there is a linear relationship between the weight of the lifted load and the normalized EMG value?

AA: The authors thank the reviewer for the comment. The RMSE of the interpolation function is now included into the manuscript. The interpolation results are reported in Section 3.3.1 and 3.3.2 as follows: (section 3.3.1) […] The goodness of the interpolation has been evaluated using the R2 and RMSE factors and reported in Table 5 for both conditions and populations, at 0o inclination, noExo and ExoON. The fitting result, obtained considering the R2 index, indicates a worst-case fitting of at least 98% on both populations, and an average RMSE of 0.23% of MVC.

(Section 3.3.2)  […] The goodness of the interpolation has been evaluated using the R2 and RMSE factors and reported in Table 5 for both conditions and populations, at 0o inclination, noExo and ExoON. The fitting result, obtained considering the R2 index, indicates a worst-case fitting of at least 97% on both populations, and an average RMSE of 0.53% of MVC.

Table 5: RMSE and R2 values of the linear interpolations

Condition

Population

R2

RMSE

noExo@ 0o
ExoON@ 0o
noExo@ 30o
ExoON@ 30o

Female
Female
Female
Female

99.92%
98.22%
97.17%
98.49%

0.115
0.344
0.797
0.302

noExo@ 0o
ExoON@ 0o
noExo@ 30o
ExoON@ 30o

Male
Male
Male
Male

99.75%
98.77%
97.44%
97.21%

0.202
0.252
0.646
0.386

Even if the interpolation goodness is always fine (higher than 97%) the linear interpolation might be not the most appropriate interpolation if considering the effective data, it is anyway a good simplification helpful for the extraction of the constant value of weight reduction (equivalent weight). It might be possible that this value is fine within a range of weight lifted and then if the lifted weight increases also these linear functions do not match the data anymore, thus a different coefficient can be used. This concept is stated in the future work of conclusions as follows: […] Moreover, the two proposed methods can be also employed a posteriori on datasets of exoskeleton performance evaluation. It is important to mentioned that the study is validated up to 15 kg of weight lifted, the same performance as EqW may not be guaranteed over a higher range of loads, so future studies will aim to evaluate the same approach covering lifting loads to the limit of the permitted weight. […]

I also have some minor concerns that are as follows:

RC2.6: Introduction- line 53, change workers to worker.

AA: Done.

RC2.7: Introduction- line 55-66, Please consider including more details regarding the results of the studies you have referenced for the 5 assessment techniques? What are some results of these assessments in the literature? This can help your argument of exoskeletons being effective.

AA: The authors thank the reviewer for the comment. We consider to clarify the concept further as follows:

[…] Recently, assessments of the effectiveness of such exoskeletons have been carried out worldwide by several groups [20,21, Pesenti2021]. Across these studies, different categories of metrics have been found useful to evaluate exoskeletons experimentally. Human biomechanics includes the analysis of movements and forces at play. For instance, estimated reduction in the compression forces and extension moments at the L5/S1 joint were found in [22 KoopmanToxiri2019]. Lifting behavior is also of interest, such as trunk inclination and movement speed. In this respect, results in different directions have been described in the literature [22 KoopmanToxiri2019] [Näf2018] [Koopman2020]. Electromyography is a technique very commonly used to assess exoskeletons, providing information on the activity of relevant muscles. Reductions in the overall activation of low-back muscles (erector spinae) in the range between 10% and 40% have been reported in several studies, but muscles in different body areas may be of interest as well [23–26]. Physiological parameters (such as heart rate, metabolic cost, breathing frequency) are indicators of global fatigue [27]. Mixed results in terms of metabolic cost can be found in the literature [Baltrusch2020] [Baltrusch2019]. Exoskeleton-related parameters include data related to the assistance delivered by the exoskeleton, such as torque and power [28], which can influence the effectiveness of a device as measured with the metrics mentioned above. Finally, questionnaires provide insights into the subjective perception of the exoskeleton, in terms of comfort, ease of use and effectiveness [29,30]. As reported in [20, Pesenti2021], a consensus on the methods and metrics for the evaluation of occupational exoskeletons is still lacking, underlining that testing conditions and the performance metrics vary across the many available studies. […]

Pesenti, M., Antonietti, A., Gandolla, M., & Pedrocchi, A. (2021). Towards a Functional Performance Validation Standard for Industrial Low-Back Exoskeletons: State of the Art Review. Sensors, 21(3), 808.

Näf, M. B., Koopman, A. S., Baltrusch, S., Rodriguez-Guerrero, C., Vanderborght, B., & Lefeber, D. (2018). Passive back support exoskeleton improves range of motion using flexible beams. Frontiers in Robotics and AI, 5, 72.

Baltrusch, S. J., Van Dieën, J. H., Koopman, A. S., Näf, M. B., Rodriguez-Guerrero, C., BabiÄŤ, J., & Houdijk, H. (2020). SPEXOR passive spinal exoskeleton decreases metabolic cost during symmetric repetitive lifting. European journal of applied physiology, 120(2), 401-412.

Baltrusch, S. J., Van Dieën, J. H., Bruijn, S. M., Koopman, A. S., Van Bennekom, C. A. M., & Houdijk, H. (2019). The effect of a passive trunk exoskeleton on metabolic costs during lifting and walking. Ergonomics.

Pesenti, M., Antonietti, A., Gandolla, M., & Pedrocchi, A. (2021). Towards a Functional Performance Validation Standard for Industrial Low-Back Exoskeletons: State of the Art Review. Sensors, 21(3), 808.

RC2.8: Introduction- line 78-79, you have claimed a poor correlation between direct measurements of physiological parameters and observational techniques such as NIOSH lifting index, OCRA, etc. Please consider including further details and citations for this claim.

AA: The authors thank the reviewer for the comment. We add the following citations:

Evaluation of Static Working Postures (ISO 11226). Author, Geneva, Switzerland.

Steven Moore, J., and Arun Garg. "The strain index: a proposed method to analyze jobs for risk of distal upper extremity disorders." American Industrial Hygiene Association Journal 56, no. 5 (1995): 443-458.

Franzblau, Alfred, Thomas J. Armstrong, Robert A. Werner, and Sheryl S. Ulin. "A cross-sectional assessment of the ACGIH TLV for hand activity level." Journal of occupational rehabilitation 15, no. 1 (2005): 57-67.

RC2.9: Introduction- line 94-98, depending on how much space or page limit allows, please consider making a stronger argument about the current barriers for the common use of exoskeletons and/or list more citations.

AA: The authors thank the reviewer for the comment. We consider to adding few lines explaining the concept further as follows: […] A number of barriers are currently hindering the adoption of exoskeletons in the workplace. Lack of consistency and comparability in the material reporting available solutions and evidence of their effectiveness, for example, contributes to generally poor awareness and makes it difficult for interested adopters to find appropriate solutions for their needs. A growing scientific literature will certainly help all stakeholders and support informed adoption. Another important barrier for widespread adoption is the lack of viable methods to quantify the benefits of occupational exoskeletons in terms of ergonomic risk. […]

RC2.10: Materials and methods- line 151, change a relatively small moment arms to relatively small moment arms

AA: Done.

RC2.11: Materials and methods- line 183- 184, change the abdominal force to the abdominal force

AA: Done.

RC2.12: Materials and methods- Figure 2B: please consider including the rx, rx1 and rx2 moment arms in the figure for better visualization.

AA: Done.

RC2.13: Materials and methods- line 196-197, Is rx2 missing from this sentence? Please clarify what rx2 refers to.

AA: Rx2 is not mentioned because is the moment arm applied on the force at the legs, now more evident in figure 2. It is out of the topic of the dissertation.

RC2.14: Materials and methods- Equations 4, please provide explanation for why Fx and Fx1 are equal. Is there a reason the torque due to the weight of the exoskeleton is not taken into account here?

AA: The authors thank the reviewer for the comment. The equations 4 and 5 have been modified.

RC2.15: Materials and methods- line 224, at the end of the line add a space between defined and between

AA: Done.

RC2.16: Materials and methods- line 227-228, did you use body weight or body mass for this equation? The text states body weight but the unit in the equation states kg which is a unit of mass. Please change one of them and make them consistent with each other.

AA: Done. It is body weight, only in appendix body weight (now body mass) in kg is used.

RC2.17: Materials and methods- line 228, please include more details about this equation. How was it developed? Please include details or more clear citations. Is it taken from reference 52?

AA: More details have been added as follows:

[…] Therefore, the coefficient of Equation 8 should be calibrated against individual maximum voluntary activation (MVA) [52]. Although there is a nonlinear relationship between the EMG measurements and muscle forces, due to cross-talking effects [53,54], in [52], by applying regression analysis, a relationship was defined between the average peak reaction moment about the L5/S1 joint and the MVA of the erectus spinae.

In this study the subject performed the MVA while lying prone on a bench and positioned so that the antero-superior iliac spines were aligned with the edge of the bench. The upper body was unsupported off the end of the bench in a horizontal trunk position. For the MVA in extension, a dynamometer was used to measure the isometric back extension force of the subject.

This research found a simple relationship between the MVA and the test subject’s body weight (BW), acting at the joint L5/S1. This is expressed in [Nm] as: MVA[Nm] = TLMAX 5/S1 = -119 + 4.78 BW[kg]. The latter equation verifies the following equilibrium: TLMAX 5/S1 = FMMAXrM - FBrBz. […]

RC2.18: Materials and methods- line 229, The torques due to abdominal force and the load are not included in this equation. Why? Please include some explanation regarding this equation or provide a clearer citation to help the reader.

AA: There is neither the abdominal pressure and nor the weight acting in the exercise of [52], the explanation of the previous point is hopefully of help also to clarify this point.

RC2.19: Materials and methods- line 311-312, the intercept can have a negative value and still have a physical meaning. Please include some explanation to why the intercept cannot be negative.

AA: Actually the intercept needs to be null and not negative. In case of negative intercept the user would be lifting a negative mass, which could be interpreted as inverse gravity.

RC2.20: Materials and methods- line 321, change remains to remained.

AA: Done.

RC2.21: Experimental assessment- line 333-336, what does this paragraph mean? The 90th percentile of what?

AA: 90th percentile is a common index used instead of the max of the EMG distribution. We calculated the 90th percentile of the envelope of the pre-processed electromyographic signal. This concept is clarified in the text.

RC2.22: Experimental assessment, line 343 and 344- change weight to mass.

AA: Done.

RC2.23: Experimental assessment- line 353, change unconsciously to subconsciously.

AA: Done.

RC2.24: Experimental assessment- line 367, not counterbalancing the order of experimental conditions (with or without the exoskeleton) can potentially affect the results. Please consider stating this as one of the limitations of the study since the results might be affected.

AA: Done.

RC2.25: Experimental assessment- line 391, on which side of the body were the EMG electrodes placed? Right or left? And why? Please specify

AA: Bilaterally, on both sides.

RC2.26: Experimental assessment- line 404-407, similar to one of the previous comments, what does this mean? 90th percentile of what?

AA: It is a standard index used very often. Text modified as follows:  […] We computed the 90th percentile of the envelope of the pre-processed EMG signal […]

RC2.27: Experimental assessment- line 416-417, what about the effect of the inclination? Please explain why the main effect of the inclination and its interaction with the other two factors was not included in the statistical model.

AA: Actually table 1 and table 3 report the statistic for both inclinations. Line 416 now include the inclination condition as well.

RC2.28: Experimental assessment- line 430, is there an L or N missing before the first reported p value representing the low load or the no load condition? Please include all four p- values even if there were not statistically significant.

AA: Done.

RC2.29: Experimental assessment- line 438, change load to loads

AA: Done.

RC2.30: Experimental assessment- Table 2, what is the tendency rate? Please include a description

AA: Tendency rate is now Trend

RC2.31: Experimental assessment- line 474, what about the p value for the M condition? Please report the value even if it is not significant.

AA: Done.

RC2.32: Experimental assessment- Table 5 and Table 6, why do you think the coefficient for the weight reduction is a larger number for females in both techniques? Please include relevant citations if needed.

AA: There is no relevant citations, it seems that females perform better.

RC2.33: Experimental assessment- line 515, change confirms to confirm.

AA: Done.

RC2.34: Discussion- line 532-534, please consider rephrasing this sentence. There are viable methods to quantify the effectiveness of exoskeletons, they may just not be simple enough, but lack of viable methods is not accurate.

AA: Now is as follows: […] remains the impossibility to quantify the ergonomic risk reduction. […]

RC2.35: Discussion- line 547, change multiply to multiplying.

AA: Done.

RC2.36: Discussion- line 551, change consider to considered

AA: Done.

RC2.37: Discussion- line 554, please consider removing the word index. CI stands for coactivation index. Therefore, CI index is redundant. There are further examples of this redundancy in this section.

AA: Done.

RC2.38: Correlation- line 566, please consider replacing the word correlation with a more suitable word. Mentioning correlation requires performing a correlation statistical analysis. Therefore, use a more accurate wording.

AA: It is now relationship.

RC2.39: Conclusion- line 600- change might be not to might not be

AA: Done.

RC2.40: Appendix A- line635 and 641, what are these question marks for?

AA: References, which should be visible.

RC2.41: Appendix A- equation A2, change trunck to trunk. There are further examples of this misspelling throughout the appendix.

AA: Done.

Round 2

Reviewer 2 Report

Thank you for submitting a revised version of your manuscript. However, some of my previous comments were not addressed in the revised version. I have included them here again. Please keep in mind the line reference numbers are from the older version of the manuscript.

Another major concern is that in the biomechanical model, the added torque on the L5/S1 joint due to the weight of the exoskeleton is not taken into consideration for the conditions in which the participant is wearing the exoskeleton.

Another major concern that I have is regarding the data interpolation model. How are we certain that there is a linear relationship between the weight of the lifted load and the normalized EMG value?

Materials and methods- line 227-228, did you use body weight or body mass for this equation? The text states body weight but the unit in the equation states kg which is a unit of mass. Please change one of them and make them consistent with each other.

Materials and methods- line 228, please include more details about this equation. How was it developed? Please include details or more clear citations. Is it taken from reference 52?

Materials and methods- line 229, The torques due to abdominal force and the load are not included in this equation. Why? Please include some explanation regarding this equation or provide a clearer citation to help the reader.

Materials and methods- line 311-312, the intercept can have a negative value and still have a physical meaning. Please include some explanation to why the intercept cannot be negative.

Author Response

R: Thank you for submitting a revised version of your manuscript. However, some of my previous comments were not addressed in the revised version. I have included them here again. Please keep in mind the line reference numbers are from the older version of the manuscript.

A: Thanks for the review and valuable suggestions that strength our manuscript. The changes associated to this round of revision have been underlined in orange.

R: Another major concern is that in the biomechanical model, the added torque on the L5/S1 joint due to the weight of the exoskeleton is not taken into consideration for the conditions in which the participant is wearing the exoskeleton.

A: The authors thank for the comment. We add the contribution in equation 6, 7, 10 and 13, in section 2.3.1 equations 14, 16, 18, 19 have been modified accordingly. In appendix A, equations A18, A19 and A20 have been modified accordingly. Also all the results of table 6 have been adjusted to include the weight of the exoskeleton in the equation of the equivalent weight. Finally figure 2 have been updated accordingly.

R: Another major concern that I have is regarding the data interpolation model. How are we certain that there is a linear relationship between the weight of the lifted load and the normalized EMG value?

A: The authors thank for the comment. The linear fit is associated with R^2 = 0.98, therefore the linear fitting is fine. The interpolations are drawn together with signals in figures 5 and 7. The authors consider the R^2 index reliable for the evaluation of the interpolation goodness. The authors also stated in conclusion that this model is valid within 15 kg. For heavier weights, further study is required.

R: Materials and methods- line 227-228, did you use body weight or body mass for this equation? The text states body weight but the unit in the equation states kg which is a unit of mass. Please change one of them and make them consistent with each other.

A: The authors thank for the comment. All body weight have been modified in body mass.

R: Materials and methods- line 228, please include more details about this equation. How was it developed? Please include details or more clear citations. Is it taken from reference 52?

A: The authors thank for the comment. Equation 8 is demonstrated in 61 and 62, and we derived the coefficient alpha by manipulation of equation: MVA [Nm]= T_{L5/S1}^{MAX} =-119+ 4.78 * BM [kg], which is presented and assessed in 63 (we add the reference close to the equation). We already included more details in the previous revision you can see the text in red at line 237.

Materials and methods- line 229, The torques due to abdominal force and the load are not included in this equation. Why? Please include some explanation regarding this equation or provide a clearer citation to help the reader.

A: The authors thank for the comment. We add few more line of explanation for the equation in line 229.

R: Materials and methods- line 311-312, the intercept can have a negative value and still have a physical meaning. Please include some explanation to why the intercept cannot be negative.

A: If negative intercept is kept, when the function is negative the equivalent weight will be also negative thus moving toward up against gravity. Otherwise, it would be possible to consider the negative intercept but whatever is negative goes at zero, in this case the function will be a composition of a null horizontal segment until the intercept with y=zero and then the function with same slope as defined in paper. With the few points that have been collected (5 10 15 kg) is not verifiable. Therefore, the authors preferred to apply the first (more conservative) solution as a simplification of the problem.  
